# Young key populations left behind: The necessity for a targeted response in Mozambique

**Makini A. S. Boothe**[1]*, **Cynthia Semá Baltazar**[1,2], **Isabel Sathane**[3], **Henry F. Raymond**[4], **Erika Fazito**[5], **Marleen Temmerman**[1,6], **Stanley Luchters**[1,7,8]

**1** Department of Public Health and Primary Care, Faculty of Medicine and Health Sciences, Ghent University, Ghent, Belgium, **2** National Institute of Health, Maputo, Mozambique, **3** National STI-HIV/AIDS Control Program, National Directorate of Public Health, Mozambique, **4** School of Public Health, Rutgers University, Piscataway, New Jersey, United States of America, **5** ICAP, Columbia University, Pretoria, South Africa, **6** Department of Obstetrics and Gynecology, Aga Khan University, Nairobi, Kenya, **7** Department of Population Health, Aga Khan University, Nairobi, Kenya, **8** Department of Epidemiology and Preventive Medicine, Monash University, Melbourne, Australia

* Makini.boothe@ugent.be, makini.boothe@gmail.com

**Data Availability Statement:** The dataset analysed for the current study are fully available from the Data Management Unit of the Mozambique National Institute of Health (INS) data repository

## Abstract

### Introduction

The first exposure to high-risk sexual and drug use behaviors often occurs during the period of youth (15–24 years old). These behaviors increase the risk of HIV infection, especially among young key populations (KP)–men how have sex with men (MSM), female sex workers (FSW), and people who inject drugs (PWID). We describe the characteristics of young KP participants in the first Biobehavioral Surveillance (BBS) surveys conducted in Mozambique and examine their risk behaviors compared to adult KP.

### Methods

Respondent-driven sampling (RDS) methodology was used to recruit KP in three major urban areas in Mozambique. RDS-weighted pooled estimates were calculated to estimate the proportion of young KP residing in each survey city. Unweighted pooled estimates of risk behaviors were calculated for each key population group and chi-square analysis assessed differences in proportions between youth (aged less than 24 years old) and older adult KP for each population group.

### Results

The majority of MSM and FSW participants were young 80.7% (95% CI: 71.5–89.9%) and 71.9% (95% CI: 71.9–79.5%), respectively, although not among PWID (18.2%, 95% CI: 13.2–23.2%). Young KP were single or never married, had a secondary education level or higher, and low employment rates. They reported lower perception of HIV risk (MSM: 72.3% vs 56.7%, p<0.001, FSW: 45.3% vs 24.4%, p<0.001), lower HIV testing uptake (MSM: 67.5% vs 72.3%, p<0.001; FSW: 63.2% vs 80.6%; p<0.001, PWID: 53.3% vs 31.2%; p = 0.001), greater underage sexual debut (MSM: 9.6% vs 4.8%, p<0.001; FSW: 35.2% vs

for researchers who meet the criteria for access to confidential data following the submission of a concept note. For information, please visit: www.ins.gov.mz or contact: secretaria@ins.gov.mz.

**Funding:** "This research has been supported by the President's Emergency Plan for AIDS Relief (PEPFAR) through the Centers for Disease Control and Prevention (CDC) under the terms of the Cooperative Agreement #U2GPS001468. The findings and conclusions in this report are those of the author(s) and do not necessarily represent the official position of the CDC."The funders had no role in study design, data collection and analysis, decision to publish, or preparation of the manuscript.

**Competing interests:** The authors have declared that no competing interests exist.

**Abbreviations:** AIS, AIDS Indicator Survey; BBS, Biobehavioral Survey; KP, Key populations; FSW, Female sex workers; HIV, Human immunodeficiency virus; MSM, Men who have sex with men (MSM); PWID, People who inject drugs (PWID); RDS, Respondent-driven sampling; RDS-A, RDS-Analyst.

22.9%, p<0.001), and greater underage initiation of injection drug use (PWID: 31.9% vs 7.0%, p<0.001). Young KP also had lower HIV prevalence compared to older KP: MSM: 3.3% vs 27.0%, p<0.001; FSW: 17.2% vs 53.7%, p<0.001; and PWID: 6.0% vs 55.0%, p<0.001. There was no significant difference in condom use across the populations.

## Conclusion

There is an immediate need for a targeted HIV response for young KP in Mozambique so that they are not left behind. Youth must be engaged in the design and implementation of interventions to ensure that low risk behaviors are sustained as they get older to prevent HIV infection.

## Introduction

The period of youth, aged 15–24 years, is a time of significant physical, mental and emotional changes characterized by exploration and inquiry that can establish behaviors that continue into adulthood [1, 2]. Although they make up 16% of the global population, youth account for one-third of new HIV infections [3, 4]. One of the core principles of the HIV response for youth is to prioritize those groups that are most vulnerable, including young key populations (KP)–men how have sex with men (MSM), female sex workers (FSW), people who inject drugs (PWID) [2]. Young KP are at high risk for HIV infection due to their high risk sexual and drug use behaviors, which often begin during this period [2, 3, 5]. In addition, young KP face social vulnerabilities specific to youth, such as power imbalances and isolation from their social support networks that may increase risk behaviors, such as early sexual debut, unprotected sex, illicit drug use and unsafe drug injection practices [6–8]. Stigma and discrimination are often heighted for this age group, who are already impacted by criminalization, but may also experience educational isolation, bullying, harassment and low quality health services [6–8].

The most recent AIDS Indicator Survey (AIS) in Mozambique estimated that youth have an HIV prevalence of 6.9% and they account for more than half of all new infections [9]. In addition, HIV testing uptake is low among sexually active youth (young women: 37.6% vs young men 18.2%) [9]. Despite this worrying situation, Mozambique does not have a comprehensive strategy to address HIV prevention among young people, although a strategy is being developed for comprehensive health promotion in schools [10]. The National Strategic Plan in Response to HIV/AIDS 2014-2019/2020 identifies adolescent girls and young women (aged 10–24) and their partners, as a priority population. Key populations are also prioritized in the National Strategic Plan and led to the development of National Guidelines for the Integration of Prevention, Care and Treatment Services for Key Populations, which set out a comprehensive package of services for key populations [11]. However, no national policy documents specifically mention young key populations nor their particular vulnerabilities [12, 13]. Finally, Mozambique's current national health information system has recently begun collecting information on KP status in HIV testing and ART services, however it is not possible to disaggregate further by age. Consequently, the national response is unable to track the HIV epidemic among young KP and cannot monitor health outcomes by age and population group, such as viral suppression or vertical transmission rates among young women.

In this context, the purpose of this secondary analysis is to describe the characteristics of youth KP aged 15–24 who participated in the first round of Bio-Behavioral Surveillance (BBS)

surveys conducted in Mozambique and to examine the differences in their risk behaviors compared to adult KP, aged 25 and older. While National strategies and interventions exist for key populations in Mozambique, more evidence is necessary in order to identify opportunities to intervene with young key populations as behaviors are being formed. Such data is needed to inform the development of innovative strategies and policies targeted specifically to this subgroup of an already vulnerable population.

## Materials and methods

### Survey design

The first round of BBS surveys in Mozambique were implemented between 2011–2014 in the country's major urban areas from the three regions: Maputo (MSM, FSW, PWID), Beira (MSM, FSW), Nampula (FSW) and Nampula/Nacala (MSM, PWID). Sampling was done using respondent-driven sampling (RDS), a quasi- probability-based peer-to-peer sampling strategy successfully used to recruit high-risk and hidden populations, whereby participants recruit their peers who meet the enrollment criteria; more information about the survey methodology has been previously published [14–16].

### Study population

Slight differences in participant eligibility criteria applied for the three target groups. Individuals were eligible for inclusion in the survey if they were aged 18 years or older (MSM, PWID) or aged 15 years or older (FSW); FSW less than 18 years old were considered emancipated minors and were therefore allowed to provide written informed consent to participate in the survey [14]. All participants had to live, work or socialize in the survey area (MSM, FSW, PWID). Specific eligibility criteria included being biologically male and having engaged in oral or anal sex with another male in the 12 months preceding the survey (MSM), being biologically female and having received money in exchange for sex from someone other than a steady partner in the six months preceding the survey (FSW) and reporting ever injecting drugs without a prescription (PWID). All participants had to present a valid referral coupon received by peer who had completed the survey. All participants provided written informed consent for the behavioral questionnaire and biological testing: HIV (MSM, FSW, PWID) and HBV/HCV (PWID). All study protocols were approved by the Mozambican National Bioethics Committee for Health, the Committee on Human Research at the University of California at San Francisco, and the Division of Global HIV/AIDS of the U.S. Centers for Disease Control and Prevention, Atlanta (USA).

### Study measures and statistical analysis

Young KP participants were defined as participants between the ages of 18–24 years old (MSM, PWID) and 15–24 years old (FSW). Analytic variables were chosen based on the literature and programmatic importance, and included demographic and health characteristics: marital status, education level, employment, HIV infection, and self-reported sexually transmitted infections (STIs) in the past 12 months; HIV-related knowledge and attitudes: experience with stigma, comprehensive HIV knowledge, HIV risk perception;, health-seeking behaviors: access of health and health services in the last 12 months, HIV test;, exposure to high-risk scenarios: physical and sexual violence, binge drinking (defined as having six or more drinks in one occasion multiple times in the past week or month); sexual risk behaviors: age of sexual debut, age of first sex work experience, multiple and concurrent sexual partners, condom use, receptive and insertive anal sex, payment or receipt of sex in exchange for money

or drugs, drug or alcohol use before last sexual encounter; and drug use: age of first illicit drug use, age at first drug injection, daily non-injection and injection drug use, illicit–non injection–drug use, injection drug use, access to clean needles, use of new syringe at last injection. Stigma was assessed by whether one believed they were refused services because of KP status (MSM, PWID). HIV knowledge was evaluated by correctly answering a standardized set of questions from the AIS [9]. Access to prevention services was a composite variable defined as having reported interaction with a peer educator, receiving free condoms, lubricants and information education and communication (IEC) materials in the last 12 months. The questionnaires for the three surveys have been previously published [17–19].

RDS-weighted pooled estimates were calculated using the aggregate estimate function of RDS-Analyst to estimate the proportion of young KP residing in each survey city. However, given the low sample size in each survey city population, the KP populations were combined across survey cities to produce unweighted aggregate estimates for further analysis. Unweighted pooled estimates were then used to conduct bivariate analysis using chi-square ($X^2$) in order to assess differences in proportions between youth and adult KP; the significance assessed at p<0.05. Descriptive analysis for aggregate age category estimates was conducted using RDS-Analyst and bivariate analyses were conducted using SAS version 9.4 (SAS Institute, Cary, NC, USA).

## Results

The majority of MSM and FSW in the survey cities were young, 80.7% (95% CI: 71.5–89.9%) and 71.9% (95%CI: 71.9–79.5%), respectively, whereas youth accounted for 18.2% (95% CI: 13.2–23.2%) of PWID, as presented in Table 1. Median age of survey participants was 21 (range: 18–59) for MSM, 21 (15–53) for FSW, and 32 (18–60) for PWID. The majority of young KP were single or never married (90.4%, 79.6% and 79.4% for MSM, FSW and PWID, respectively). Among young PWID, the majority were male (93.5%). For all three groups, more than two-thirds reported secondary education or higher (MSM: 84.5%, FSW: 70.7%, PWID: 69.6%). About half of MSM participants reported employment (53.1%), compared to one-fifth of FSW who reported work aside from sex work (20.3%); employment was not included in the survey instrument for PWID. Among male survey participants, 34.9% of MSM and 19.8% of PWID were uncircumcised; 59.7% of young FSW reported ever being pregnant. HIV prevalence was estimated at 3.3% for young MSM, 17.2% for young FSW and 6.0% for young PWID. Self-reported sexually transmitted infection (STI) was lower for MSM (11.2%) compared to both FSW (31.4%) and PWID (34.8%).

**Table 1. Aggregate RDS-weighted estimates of adolescents and young people among MSM, FSW and PWID, Mozambique, 2011–2014.**

| | Maputo | | | Beira* | | | Nampula/Nacala | | | Total | | |
|---|---|---|---|---|---|---|---|---|---|---|---|---|
| | n/N: Crude | %: Crude | RDS-weighted (95% CI) | n/N: Crude | %: Crude | RDS-weighted (95% CI) | n/N: Crude | %: Crude | RDS-weighted (95% CI) | n/N: Crude | %: Crude | RDS-weighted (95% CI) |
| MSM | 385/496 | 77.6 | 79.6 (65.4–93.9) | 456/583 | 78.2 | 79.2 (73.3–85.0) | 293/353 | 83.0 | 85.5 (80.6–90.5) | 1134/1432 | 79.2 | 80.7 (71.5–89.9) |
| FSW | 238/400 | 59.5 | 65.0 (50.2–79.7) | 317/411 | 77.1 | 78.8 (72.6–85.0) | 333/429 | 77.6 | 78.6 (72.7–84.4) | 888/1240 | 71.6 | 71.9 (64.3–79.5) |
| PWID | 37/353 | 10.5 | 11.9 (6.6–17.1) | | | | 55/139 | 39.6 | 38.5 (26.0–51.0) | 92/492 | 18.7 | 18.2 (13.2–23.2) |

Note: Men who have sex with men (MSM), age 18–24; Female sex workers (FSW), age 15–24; People who inject drugs (PWID), age 18–24

* The BBS among PWID was not conducted in Beira

## Young MSM

Results from the unweighted pooled bivariate estimates of risk factors among young MSM are presented in Table 2. Younger MSM were more likely to be single compared to older MSM (90.4% vs 59.3%, p<0.001) and to have a secondary education level or higher (84.5% vs 79.1%, p = 0.028); however, there were lower rates of current employment (53.1% vs 85.5%, p<0.001). Compared to adult MSM, young MSM reported greater stigma (9.9% vs 5.7%, p = 0.026) and lower health-seeking behaviors in the last 12 months (p = 0.001). More youth had a low perception of their HIV risk (72.3% vs 56.7%, p<0.001), and more reported never having an HIV test (p<0.001); there was no difference found in comprehensive HIV knowledge between the two groups. HIV prevalence were lower among youth 3.3% vs 27.0% (p<0.001), and the same dynamic was observed for self-reported STIs: 11.2% vs 20.5% (p<0.001). Young MSM also reported less binge drinking behaviors (31.6% vs 43.8%, p<0.001) and less illicit drug use in the past 12 months (8.9% vs 12.8%, p = 0.045). Regarding sexual risk behaviors, more young MSM reported their first anal sexual encounter with a man having occurred before the age of 15 years old: 9.6% vs 4.8% (p = 0.023); they also reported less experiences of paying or

**Table 2. Socio-demographic and behavioral risk factors of young (18–24 year old) and older adult (25+) Men who have sex with men (MSM) participants, Mozambique 2012.**

| | 18–24 | | 25+ | | p-value |
| --- | --- | --- | --- | --- | --- |
| | n | % | n | % | |
| Single or never married | 1022 | 90.4 | 176 | 59.3 | **<0.001** |
| Secondary education level or higher | 956 | 84.5 | 235 | 79.1 | **0.028** |
| Currently employed | 601 | 53.1 | 254 | 85.5 | **<0.001** |
| Uncircumcised | 395 | 34.9 | 115 | 38.7 | 0.221 |
| Experienced stigma in the last 12 months | 112 | 9.9 | 17 | 5.7 | **0.026** |
| Did not seek health services in the last 12 months | 604 | 53.4 | 126 | 42.4 | **0.001** |
| Low perception of HIV risk[a] | 790 | 72.3 | 157 | 56.7 | **<0.001**[a] |
| Lack of comprehensive HIV knowledge | 529 | 46.7 | 134 | 45.1 | 0.620 |
| Never had HIV test | 481 | 42.5 | 82 | 27.7 | **<0.001** |
| HIV infection | 36 | 3.3 | 78 | 27.0 | **<0.001** |
| Self-reported STI | 127 | 11.2 | 61 | 20.5 | **<0.001** |
| No access to comprehensive prevention services[b] | 760 | 67.1 | 202 | 68.0 | 0.775 |
| Binge drinking[c] | 347 | 31.6 | 126 | 43.8 | **<0.001** |
| Illicit drug use in the last 12 months | 101 | 8.9 | 38 | 12.8 | **0.045** |
| Physical Violence in the last 12 months | 43 | 3.8 | 8 | 2.7 | 0.366 |
| Sexual Violence in the last 12 months | 14 | 1.2 | 4 | 1.4 | 0.776 |
| Less than 15 years old at first anal sexual experience | 108 | 9.6 | 14 | 4.8 | **0.023** |
| 2+ male anal sex partners[d] | 494 | 45.6 | 138 | 48.3 | 0.426 |
| Concurrent male and female sexual partner in the last 12 months | 560 | 49.5 | 176 | 59.3 | **0.003** |
| Receptive anal sex in the last 12 months | 411 | 36.3 | 112 | 37.7 | 0.655 |
| Insertive anal sex in the last 12 months | 967 | 85.4 | 252 | 84.9 | 0.803 |
| No Condom use at last sexual encounter | 304 | 27.0 | 96 | 32.5 | 0.060 |
| Paid or received sex in exchange for money in the last 12 months | 484 | 42.8 | 152 | 51.2 | **0.010** |

*Significance level assessed at p<0.05

[a] Analysis excludes those with knowledge of HIV-positive status

[b] Contact with a peer educator and received free condoms, lubricants, and Information education and communication (IEC) materials

[c] Binge drinking defined as having six or more drinks in one occasion multiple times in the past week or month.

[d] Analysis only includes MSM who reported anal sex in last 12 months.

receiving money in exchange for sex (42.8% vs 51.2%, p = 0.01). The results did not show a difference in circumcision rates, access to comprehensive prevention services, physical or sexual violence, number of anal sex male partners, receptive and insertive anal sex, or non-condom use.

## Young FSW

Table 3 presents the socio-demographic and behavioral risk factors of adolescent and young FSW participants compared to the older adult participants. Among FSW participants, more young FSW reported being single or never married (79.6% vs 26.5%, p<0.001) and having a secondary education level or higher (70.7% vs 43.6%, <0.001); however, less reported having other work outside of sex work (20.3% vs 32.5%; p<0.001). More youth compared to adults had a low perception of their HIV risk (45.3% vs 24.4%, p<0.001) and more reported never having had an HIV test (36.8% vs 19.4%, p<0.001). Unadjusted pooled HIV prevalence was lower among young FSW (17.2% vs 53.7%, p<0.001), although there was no significant difference of self-reported STI infection between the two groups (31.4% vs 33.2%, p = 0.536). More youth than adults reported not having access to comprehensive prevention services (86.5% vs 75.2%, p<0.001) and more reported not having sought health services in the last 12 months (64.0% vs 51.3%, p<0.001). Youth had less experiences of binge drinking (24.4% vs 33.7%, p = 0.001). Young FSW participants experienced more physical and sexual violence, 15.3% vs 10.6% (p = 0.031) and 12.5% vs 7.7% (p = 0.016), respectively. More young FSW reported both

**Table 3. Socio-demographic and behavioral risk factors of young (15–24 year old) and older adult (25 years and older) Female Sex Workers (FSW) participants, Mozambique 2011–2012.**

|  | 18–24 | | 25+ | | p-value |
|---|---|---|---|---|---|
|  | n | % | n | % |  |
| Single or never married | 705 | 79.6 | 93 | 26.5 | **<0.001** |
| Secondary education level or higher | 626 | 70.7 | 153 | 43.6 | **<0.001** |
| Work aside from sex work | 180 | 20.3 | 114 | 32.5 | **<0.001** |
| Ever Pregnant | 529 | 59.71 | 327 | 93.16 | **<0.001** |
| Low perception of HIV risk | 361 | 45.3 | 79 | 24.4 | **<0.001** |
| Lack of comprehensive HIV knowledge | 414 | 46.7 | 143 | 40.7 | 0.056 |
| Never had HIV test | 326 | 36.8 | 68 | 19.4 | **<0.001** |
| HIV infection | 152 | 17.2 | 189 | 53.7 | **<0.001** |
| Self-reported STI, last 12 months | 279 | 31.4 | 117 | 33.2 | 0.536 |
| No access to comprehensive prevention services[a] | 766 | 86.5 | 264 | 75.2 | **<0.001** |
| Did not seek health services in the last 12 months | 567 | 64.0 | 180 | 51.3 | **<0.001** |
| Binge drinking[b] | 216 | 24.4 | 118 | 33.7 | **0.001** |
| Illicit drug use in the last 12 months | 17 | 1.9 | 7 | 2.0 | 0.931 |
| Physical Violence in the last 12 months | 135 | 15.3 | 37 | 10.6 | **0.031** |
| Sexual Violence in the last 12 months | 111 | 12.5 | 27 | 7.7 | **0.016** |
| Less than 15 years old at first sexual experience | 310 | 35.2 | 78 | 22.9 | **<0.001** |
| Less than 15 years old at first sex work experience | 111 | 12.6 | 15 | 4.4 | **<0.001** |
| Ever had anal sex | 180 | 20.3 | 90 | 25.6 | **0.041** |
| No Condom use at last sexual encounter, with client | 217 | 24.5 | 99 | 28.3 | 0.172 |
| No Condom use at last sexual encounter, with non-client partner[c] | 156 | 45.1 | 58 | 55.8 | 0.056 |

[a] Contact with a peer educator and received free condoms, lubricants, and Information education and communication (IEC) materials.

[b] Binge drinking defined as having six or more drinks in one occasion multiple times in the past week or month.

[c] Analysis only includes FSW with non-client partners.

sexual initiation and first sex work experience before the age of 15 years old, 35.2% vs 22.9% (p<0.001) and 12.6% vs 4.4% (p<0.001), respectively. There was no reported difference in comprehensive HIV knowledge, illicit drug use, self-reported STI, and condom use with last partner or non-client partner.

## Young PWID

As presented in Table 4, more young PWID also reported being single or never married compared to their adult counterparts (79.4% vs 54.0%, p<0.001) and more also reported having a secondary education level or higher (69.6% vs 54.4%, p = 0.008); less young PWID reported a history of arrest (44.6% vs 71.8%, p<0.001). Never having had an HIV test was reported by more young PWID compared to older PWID (46.7% vs 28.8%, p = 0.001); they had much lower HIV prevalence (6.0% vs 55.0%, p<0.001), although there was no difference in HIV risk perception (34.3% vs 40.6%, p = 0.340). Self-reported STI infection was greater among youth PWID compared to older adults (34.8% vs 21.3%, p = 0.006). Less young PWID reported not having access to comprehensive prevention services (76.1% vs 86.0%, p = 0.019). More young PWID reported having multiple sexual partners in the past year (77.3% vs 53.7%, p<0.001), although there was no difference in condom use at last sexual encounter (47.8% vs 42.6%,

**Table 4. Socio-demographic and behavioral risk factors of young (15–24 year old) and older adult (25 years and older) People who inject drugs (PWID) participants, (n = 492), Mozambique 2014.**

| | 18–24 | | 25+ | | p-value |
|---|---|---|---|---|---|
| | n | % | n | % | |
| Male | 86 | 93.5 | 381 | 95.3 | 0.485 |
| Single or never married | 73 | 79.4 | 216 | 54.0 | <0.001 |
| Secondary education level or higher | 64 | 69.6 | 217 | 54.4 | 0.008 |
| Uncircumcised | 17 | 19.8 | 143 | 37.5 | 0.003 |
| History of arrest | 41 | 44.6 | 287 | 71.8 | <0.001 |
| Low perception of HIV risk[a] | 25 | 34.3 | 84 | 40.6 | 0.340 |
| Never HIV Test | 43 | 46.7 | 115 | 28.8 | 0.001 |
| HIV infection | 5 | 6.0 | 199 | 55.0 | <0.001 |
| Self-reported STI | 32 | 34.8 | 85 | 21.3 | 0.006 |
| Did not seek health services | 54 | 58.7 | 253 | 63.3 | 0.416 |
| No access to comprehensive prevention services[b] | 70 | 76.1 | 344 | 86.0 | 0.019 |
| Physical Violence | 18 | 19.6 | 59 | 14.8 | 0.252 |
| Sexual Violence | 3 | 3.3 | 3 | 0.8 | 0.082 |
| Experienced stigma | 17 | 19.3 | 67 | 17.5 | 0.687 |
| 2+ sexual partners, last 12 months | 68 | 77.3 | 211 | 53.7 | <0.001 |
| No condom use at last sexual encounter | 44 | 47.8 | 170 | 42.6 | 0.627 |
| Drugs or alcohol use before last sexual encounter | 13 | 15.1 | 76 | 19.8 | 0.415 |
| Received drugs in exchange for sex | 14 | 15.2 | 53 | 13.3 | 0.626 |
| Less than 18 years old at first drug use | 56 | 61.5 | 136 | 34.7 | <0.001 |
| Less than 18 years old at first injection use | 29 | 31.9 | 27 | 7.0 | <0.001 |
| Daily drug use (including non-injection drugs) | 40 | 12.2 | 289 | 73.9 | <0.001 |
| Daily injection drug use | 27 | 29.4 | 248 | 62.0 | <0.001 |
| No access to new syringes | 16 | 17.4 | 50 | 12.5 | 0.218 |
| No use of new syringe at last injection | 26 | 30.2 | 148 | 39.1 | 0.127 |

[a] Analysis excludes those with knowledge of HIV-positive status

[b] Contact with a peer educator and received free condoms, lubricants, and Information education and communication (IEC) materials.

p = 0.627). More young PWID reported first illicit drug use experience and first injection drug experience before the age of 18 years old, 61.5% vs 34.7% (p<0.001) and 31.9% vs 7.0% (p<0.001), respectively. They reported less daily illicit (non-injection) drug use (12.2% vs 73.9%, p<0.001) and less daily injection use (12.2% vs 73.9%, p<0.001). There was no significant difference between access to new syringes and use of new syringes at last injection.

## Discussion

About a quarter of the adult population in Mozambique is between the ages of 15–24 years old, although the overwhelming majority of MSM and FSW in the surveyed cities are estimated to be youth: 81% and 72%, respectively [20, 21]. The younger profile of MSM is consistent with what has been observed in the literature [22–25], while the age distribution of FSW varies by context [26–29]. The proportion of young PWID is consistent with the youth demographic in the general population [21]; the older age profile of PWID is also consistent with literature from the region [30–33]. Unweighted pooled estimates presented across the three populations demonstrate that youth were generally single or never married and had higher education levels, which is consistent with the general population [9]. Young MSM and FSW reported greater unemployment than the adult KP population, which is similar to the general population [9].

Stigma was only assessed for MSM and PWID and show greater experiences among young MSM; there was also no difference in stigma among PWID. Other studies and systematic reviews point to the role of stigma on risk behaviors and low health-seeking behaviors, where youth fear discrimination from health care workers, family, community members, teachers and classmates [34, 35].

Across all three populations, young KP had a lower perception of their HIV risk and lower comprehensive knowledge compared to their adult counterparts. Compared to the general population, comprehensive knowledge of HIV among young KP was higher than that among youth in the general population aged 15–24 (MSM: 53.3%, FSW: 53.3% compared to Young men: 30.8% and Young women: 30.2%) [9]. Perhaps not surprisingly, although young KP reported less HIV testing than the adults across the three populations, they reported greater HIV testing than their counterparts in the general population, where close to three-fifths of young women reported a previous HIV test and less than a third of young men [9]. Both comprehensive knowledge and HIV testing demonstrate that young KP are more aware of their risk compared to their counterparts in the general population. Estimated HIV prevalence among young MSM (3.3%) is similar to their counterparts in the general population (3.2%), however young FSW have almost double HIV prevalence compared to women in their same age group: 9.8% vs 17.2%. HIV prevalence among youth PWID (6.0%)—the majority of whom were male—was higher than among male youth in the general population (3.2%) [9].

Alarmingly high proportions of young KP reported not having access to prevention services in the last 12 months, although only statistically significant among PWID. This finding, coupled with low comprehensive HIV knowledge and low testing uptake, highlight that young KP have not been empowered to take charge of their own health of HIV prevention. This is particularly worrying for FSW, close to two-thirds of whom report ever being pregnant and risk transmitting HIV vertically to their children [36]. This calls for enhance youth-specific HIV interventions, using peer educators, mobile technology and social media [5, 37]. Other clinical innovations, such as HIV self-testing (HIVST), may be important for this age group, which are needed to address the low testing rates across all three KP groups. For example, a study in Uganda found that MSM preferred HIVST than traditional HIV testing strategies with peers

or at hot spots, drop-in centers, private pharmacies and MSM service providers; this was also reinforced by research coming from South Africa [38, 39].

Self-reported STI infection in the last 12 months was higher among young key populations than that reported by the general population (MSM: 11.2%, FSW: 31.4%, PWID: 34.8% vs 4.0% for young women and 5.5% for young men) [9]. Given the mode of transmission of HIV, the prevention and diagnosis of STIs among young KP must be integrated into any HIV prevention and treatment services. While young MSM reported less STIs than their adult counterparts, young PWID had higher self-reported STIs. Although there was no difference in self-reported STI infection between younger and older adult FSW, at least one-third of each group reported STI infection, thus confirming the importance of this health issue for that population. While alarming, estimates of self-reported STI infection defined by symptoms or previous diagnosis in the 12 months preceding the survey are likely underestimated. WHO guidelines currently call for syndromic case diagnosis, as measured in the surveys, however, given that most STIs are asymptomatic, the absence of laboratory testing excluded asymptomatic cases, potentially underestimating STI prevalence [40, 41].

Young PWID have more problematic sexual risk behaviors than their older counterparts, where greater proportions report multiple sexual partners, drug or alcohol use before their last sexual encounter and greater self-reported STIs. These risk behaviors illustrate the compounded risk pathways of HIV transmission due to both sexual risk and injection drug use behaviors. Of note, KP across all three populations report higher condom use at their last sexual encounter compared to the general population (MSM: 73.3%, FSW: 75.5% with client and 54.9% with non-client partner, PWID: 52.2% vs young women: 42.0% and young men: 39%), highlighting the impact of condom promotion interventions among these high-risk groups.

As observed in other studies, the results display that risk behaviors begin at younger ages, such as earlier sexual debut (MSM, FSW) and earlier drug use and injection drug use experiences among PWID [22, 34]. Prolonged exposure can eventually lead to adverse health outcomes such as HIV and STI infection, emphasizing an urgent need to prevent and/or reinforce healthy preventative behaviors over time into adulthood. That younger PWID report daily injection use at lower rates than adults represents a prime opportunity for intervention before more adverse injection behaviors are adopted.

Any analyses of youth KP must acknowledge that they are not a homogenous group and therefore interventions must also address the intersectionality of risk profiles. As observed, 42.8% of young MSM reported paying or receiving money in exchange for sex and 15.2% of young PWID reported receiving sex in exchange of drugs; 8.9% of young MSM were (non-injection) drug users. Evidence of overlapping risk profiles has been explored in different contexts, representing sub-groups within already vulnerable populations who may have a higher risk of HIV infection than those with one risk behavior or identity [6, 31, 41–44]. Formative research is required to explore health-seeking behaviors of young key populations and their preference for accessing services whether through youth-friendly services, key populations friendly services or services specifically defined for young key populations. Such an inquiry could also explore the implications of compounded structural, cultural and social barriers on use of services. In addition, interventions should be characterized by meaningful engagement with and leadership by community organizations, particularly those with young peer educators.

Various social and structural barriers contribute to the heighted vulnerabilities of young KP. For example, although young PWID report lower experience with arrest compared to adults, close to half reported a history of arrest thus underscoring the criminalization of addiction. Similarly, young FSW reported higher levels of sexual and physical violence in the past 12 months compared to their older counterparts, likely due to unequal power dynamics and

patriarchal social structures. These particular vulnerabilities could be addressed through empowerment programs of key populations on human rights and legal provisions. Social networks could also be leveraged to provide safety, protection and advocacy for female sex workers experiencing violence. Finally, structural barriers could be confronted through partnerships with police and other law enforcement agencies that promote accountability.

Thus, both behavioral and structural interventions are of paramount importance and the participatory engagement of youth in the design and implementation of programming for a targeted response cannot be ignored [2, 25, 39, 45–47]. UNAIDS outlines the importance of capacity building initiatives of youth-led organizations and associations to ensure their ability to mobilize and advocate for their peers [3]. This can include practical skills, such as capacity building in grant development, human and financial resource management and systems for monitoring and evaluating the reach and impact of programming. In addition, interventions must address the intra- and interpersonal factors contributing to high risk behaviors in KP youth such as low self-esteem, loneliness and perceived lack of social support [48]. Structural interventions must also address the particular vulnerabilities of young KP such as keeping girls in school, creating employment opportunities, the decriminalization of drug addiction, and should promote human rights [3]. These approaches must be include a coordinated response with civil society organizations and the various government sectors responsible for youth programming, most notably Health, Education and Human Development, and Youth and Sports.

Finally, it is not currently possible to track and analyze the HIV epidemic among the adolescent population in general, and young KP specifically, because of limited health information systems. Therefore the scope of young KP engagement in prevention, care and treatment services, and subsequent health outcomes such as viral suppression and vertical transmission among young women, is unknown. However, as the youth population continues to grow, so too does the risk of HIV infection among young KP, if targeted efforts are not urgently adopted. As an illustration, population growth among youth aged 15–24 in Mozambique resulted in an additional 53,000 new infections between 2010–2017 [36].

Although this is the first analysis of young KP in Mozambique, there are several limitations to be discussed. First, this analysis is subject to the general limitations of RDS surveys such as selection bias in peer-referral sampling methods, recall bias, and social desirability bias. Next, given the small sample size, the bivariate analysis was conducted on unweighted aggregate estimates, which removed social networks and chains, and therefore the results may not be generalizable to KP and simply represent the survey participants. In addition, the study was not powered to compare youth and adult KP so true associations may not have been captured. Finally, the ability to compare across KP groups was limited by the survey measures. Although the survey instruments were largely consistent across the three populations, some key variables were missing such as stigma estimates for FSW and employment, comprehensive HIV knowledge, binge drinking, and age of sexual debut for PWID.

Despite these limitations, this is the only available study examining the sexual risk and drug use behaviors of young key populations in Mozambique and reinforces the importance of early interventions in order to promote lifelong health status.

## Conclusion

This analysis points to the need for targeted strategic participatory approaches to address the specific risk profile of young key populations, specifically for FSW and PWID. Future analyses could the engagement of young key populations in prevention, care and treatment services such as PrEP, ART and viral load testing. Mozambique is currently developing a Prevention Roadmap, which will become the guiding document for the design, implementation and

monitoring of prevention services, including those for key populations. Such a strategic document would be strengthened by including specific attention on young key populations. Given the HIV epidemic in Mozambique, and the large demographic of youth, issues in adolescent and youth health, especially among those most at risk for HIV infection, must be addressed to guarantee that these groups are not left behind in the HIV response and are able to maintain healthy behaviors into adulthood.

## Acknowledgments

The study team acknowledges the immense contributions of the Mozambican BBS Technical Working Group, and all who have contributed to the successful implementation of the MSM, FSW and PWID surveys in Mozambique.

## Author Contributions

**Conceptualization:** Makini A. S. Boothe.

**Data curation:** Isabel Sathane.

**Formal analysis:** Makini A. S. Boothe.

**Investigation:** Cynthia Semá Baltazar, Henry F. Raymond.

**Methodology:** Isabel Sathane.

**Project administration:** Cynthia Semá Baltazar, Isabel Sathane.

**Supervision:** Henry F. Raymond, Erika Fazito, Marleen Temmerman, Stanley Luchters.

**Validation:** Isabel Sathane, Henry F. Raymond.

**Writing – original draft:** Makini A. S. Boothe.

**Writing – review & editing:** Cynthia Semá Baltazar, Henry F. Raymond, Erika Fazito, Marleen Temmerman, Stanley Luchters.

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
