## [Decision Letter · Decision Letter 0]

4 Oct 2021

PONE-D-21-05957Young key populations left behind: The necessity for a targeted response in MozambiquePLOS ONE

Dear Dr. Boothe,

Thank you for submitting your manuscript to PLOS ONE. After careful consideration, we feel that it has merit but does not fully meet PLOS ONE’s publication criteria as it currently stands. Therefore, we invite you to submit a revised version of the manuscript that addresses the points raised during the review process.

We look forward to receiving your revised manuscript.

Kind regards,

Lorena Verduci

Academic Editor

PLOS ONE

Journal Requirements:

Additional Editor Comments (if provided):

Reviewers' comments:

Reviewer's Responses to Questions

**Comments to the Author**

1. Is the manuscript technically sound, and do the data support the conclusions?

Reviewer #1: Yes

Reviewer #2: Yes

2. Has the statistical analysis been performed appropriately and rigorously? 

Reviewer #1: Yes

Reviewer #2: I Don't Know

3. Have the authors made all data underlying the findings in their manuscript fully available?

Reviewer #1: Yes

Reviewer #2: Yes

4. Is the manuscript presented in an intelligible fashion and written in standard English?

Reviewer #1: Yes

Reviewer #2: Yes

5. Review Comments to the Author

Reviewer #1: This is a useful analysis of the experiences of young KP groups that is often neglected in Southern Africa.

I have only minor comments on the paper:

1) It is not until the discussion that it is stated that self reported STIs related to over the last 12 months – this should be in the methods. There should be some reflection of the fact that this will be an underestimate as only relates to symptomatic STIs.

2) What was the time period of reported ’binge drinking’?

3) The rates of MSM reporting payment for sex is high. It would be valauble to comment in the discussion about the need for interventions for MSM involved in sex work which is rarely highlighted in Southern Africa.

4) What about the role of PrEP for KPs?

5) What about estimates of KPs on ART?

6) The levels of violence expereinece by FSWs is also very high – and reference could be made to interventions to reduce client based and partner based vilence about FSWs in the discussion

it is a nice paper and good to see results from Mozambique.

Reviewer #2: This paper examines HIV risk behaviours among young people aged 15-24 years in three major urban areas in Mozambique. The study focuses on injecting drug users, female sex workers and men who have sex with men. Recruitment through respondent-driven sampling which is appropriate for hard to reach populations and where population data is not available. In the analysis risk behaviours among these young people were compared to adult risk populations.

The paper is well-written and I have very few comments to make overall. I am not the right person to assess the RDS – but see that the study has publications that have previously been through review and hope that another reviewer with appropriate statistical skills would be able to comment on that element.

An overarching query about the paper as a whole. Given that there is no KP HIV prevention strategy in Mozambique, I was wondering about the value of comparing young KP with older KP. If Mozambique has already defined young people as a priority population – would an analysis that compared these KP with ‘other youth’ not be more valuable in making a case for specific focus on KP within the youth priority population? It would be useful to have some more information about the rationale for this particular analysis, as this is not currently completely clear, particularly given that much of the discussion focuses on comparison of the youth KP with the general youth population, which appears to have already been done. How will this particular analysis assist in the development of strategy to assist the Mozambican government act to prevent HIV? I think that it would also be useful to provide some context about how HIV prevention is structured in Mozambique and how these findings might suggest a need to change this i.e are young KP more likely to get assistance in a KP-orientated service or a youth-orientated service? Or do neither exist?

I wonder whether some of my questions might be answered if the information about inability to disaggregate KP by age in the data was discussed in the Introduction?

Abstract: Assume that the comparisons of the young KP with older populations should all be percentages. It would be worth adding these in for greater clarity.

Page 15. Line 222: change to ‘higher education levels’

Page 19, line 312: review this line. The ‘however’ suggests that the second half of the sentence contradicts or is different to the first and this is not the case in this instance.

6. PLOS authors have the option to publish the peer review history of their article (what does this mean?). If published, this will include your full peer review and any attached files.

Reviewer #1: **Yes: **Mitzy Gafos

Reviewer #2: No

---

## [Author Response · Author response to Decision Letter 0]

18 Nov 2021

Reviewer #1: 

It is not until the discussion that it is stated that self-reported STIs related to over the last 12 months – this should be in the methods. There should be some reflection of the fact that this will be an underestimate as only relates to symptomatic STIs.

Thank you for this observation. The methodology has been revised as follows: 

Analytic variables were chosen based on the literature and programmatic importance, and included demographic and health characteristics: marital status, education level, employment, HIV infection, and self-reported sexually transmitted infections (STIs) in the past 12 months;

The discussion section provides a presentation of the limitations of symptomatic STI, as follows: 

While alarming, estimates of self-reported STI infection defined by symptoms or previous diagnosis in the 12 months preceding the survey are likely underestimated. WHO guidelines currently call for syndromic case diagnosis, as measured in the surveys, however, given that most STIs are asymptomatic, the absence of laboratory testing excluded asymptomatic cases, potentially underestimating STI prevalence.

What was the time period of reported ’binge drinking’?

Binge drinking was defined as having six or more drinks in one occasion multiple times in the past week or month. The text and tables have been revised accordingly.

The rates of MSM reporting payment for sex is high. It would be valuable to comment in the discussion about the need for interventions for MSM involved in sex work which is rarely highlighted in Southern Africa.

Thank you for this valuable comment. As you correctly point out, participation in transactional sex was high among MSM - whether as clients or sex workers - with a significant difference observed by age (42.8% vs 51.2%, p=0.01). A discussion for MSM sex work has been included as follows:

Any analyses of youth KP must acknowledge that they are not a homogenous group and therefore interventions must also address the intersectionality of risk profiles. As observed, 42.8% of young MSM reported paying or receiving money in exchange for sex and 15.2% of young PWID reported receiving sex in exchange of drugs; 8.9% of young MSM were (non-injection) drug users. Evidence of overlapping risk profiles has been explored in different contexts, representing sub-groups within already vulnerable populations who may have a higher risk of HIV infection than those with one risk behavior or identity (6,30,40–43). A person-centered approach to interventions is required that takes into consideration the entirety of risk behaviors as well as the compounded impact of social, cultural, and structural barriers on access and use of prevention and treatment services. In addition, interventions should be characterized by meaningful engagement with and leadership by community organizations, particularly those with young peer educators. 

What about the role of PrEP for KPs?

PrEP was not assessed for KPs at the time of survey implementation (2011-2014) since it was not yet available in country; in addition, PrEP expansion in Mozambique was initiated in 2021. The text has been revised as follows to address opportunities for future research: 

Future analyses could the engagement of young key populations in prevention, care and treatment services such as PrEP, ART and viral load testing. 

What about estimates of KPs on ART?

ART coverage was explored in a separate manuscript titled “Low engagement in HIV services and progress through the treatment cascade among key populations living with HIV in Mozambique, 2021” (available here). This analysis showed that engagement of KP in the HIV testing and treatment cascade at the time of survey implementation was drastically low: 3.5% among MSM, 11.7% among FSW, and 29.4% among PWID. Given the low engagement of key populations, a meaningful discussion about differences by age would be limited. 

However, ART has been included in the discussion as an area of future inquiry, as follows: 

Future analyses could the engagement of young key populations in prevention, care and treatment services such as PrEP, ART and viral load testing. 

The levels of violence experienced by FSWs is also very high – and reference could be made to interventions to reduce client based and partner based violence about FSWs in the discussion

Thank you for this recommendation, a discussion of violence prevention interventions for sex workers has been presented as follows: 

Various social and structural barriers contribute to the heighted vulnerabilities of young KP. For example, although young PWID report lower experience with arrest compared to adults, close to half reported a history of arrest thus underscoring the criminalization of addiction. Similarly, young FSW reported higher levels of sexual and physical violence in the past 12 months compared to their older counterparts, likely due to unequal power dynamics and patriarchal social structures. These particular vulnerabilities could be addressed through empowerment programs of key populations on human rights and legal provisions. Social networks could also be leveraged to provide safety, protection and advocacy for female sex workers experiencing violence. Finally, structural barriers could be confronted through partnerships with police and other law enforcement agencies that promote accountability. 

Reviewer #2: 

Given that there is no KP HIV prevention strategy in Mozambique, I was wondering about the value of comparing young KP with older KP. If Mozambique has already defined young people as a priority population – would an analysis that compared these KP with ‘other youth’ not be more valuable in making a case for specific focus on KP within the youth priority population? It would be useful to have some more information about the rationale for this particular analysis, as this is not currently completely clear, particularly given that much of the discussion focuses on comparison of the youth KP with the general youth population, which appears to have already been done. How will this particular analysis assist in the development of strategy to assist the Mozambican government act to prevent HIV? I think that it would also be useful to provide some context about how HIV prevention is structured in Mozambique and how these findings might suggest a need to change this i.e are young KP more likely to get assistance in a KP-orientated service or a youth-orientated service? Or do neither exist?

Thank you for this rich feedback. 

The statistical analyses presented in the manuscript compare adult KP to young KP in an effort to tailor interventions. Of final note, there has not been a comprehensive comparison of young key populations to youth in the general population, aside from the proportions in the discussion section, as the survey methodologies are different (ex: the current analysis looks like behaviors among survey participants across a few cities while the youth from the AIDS Indicator Survey produces estimates for generalizable results from a national level

Based on this comment, the following passages have been included in the manuscript: 

Introduction 

The National Strategic Plan in Response to HIV/AIDS 2014-2019/2020 (NSP) identifies adolescent girls and young women (aged 10-24) and their partners, as a priority population. Key populations are also prioritized in the National Strategic Plan, which led to the development of National Guidelines for the Integration of Prevention, Care and Treatment Services for Key Populations, which set out a comprehensive package of services for key populations. However, no national policy documents specifically mention young key populations nor their particular vulnerabilities. 

While National strategies and interventions exist for key populations in Mozambique, more evidence is necessary in order to identify opportunities to intervene with young key populations as behaviors are being formed. Such data is needed to inform the development of innovative strategies and policies targeted specifically to this sub-group of an already vulnerable population.

Discussion

Formative research is required to explore health-seeking behaviors of young key populations and their preference for accessing services whether through youth-friendly services, key populations friendly services or services specifically defined for young key populations. Such an inquiry could also explore the implications of compounded structural, cultural and social barriers on use of services. 

Conclusion

Mozambique is currently developing a Prevention Roadmap, which will become the guiding document for the design, implementation and monitoring of prevention services, including those for key populations. Such a strategic document would be strengthened by including a specific attention on young key populations. Given the HIV epidemic in Mozambique, and the large demographic of youth, issues in adolescent and youth health, especially among those most at risk for HIV infection, must be addressed to guarantee that these groups are not left behind in the HIV response and are able to maintain healthy behaviors into adulthood. 

I wonder whether some of my questions might be answered if the information about inability to disaggregate KP by age in the data was discussed in the Introduction?

Thank you, the description of the Mozambique Health Information System has been moved to the Introduction section and been edited for clarity:

Mozambique’s current national health information system has recently begun collecting information on KP status in HIV testing and ART services, however it is not possible to disaggregate further by age. Consequently, the national response is unable to track the HIV epidemic among adolescents and cannot monitor health outcomes by age and population group, such as viral suppression or vertical transmission rates among young women. 

The Discussion section has also been updated as follows: 

Finally, it is not currently possible to track and analyze the HIV epidemic among the adolescent population in general, and young KP specifically, because of limited health information systems. Therefore the scope of young KP engagement in prevention, care and treatment services, and subsequent health outcomes such as viral suppression and vertical transmission among young women, is unknown

Abstract: Assume that the comparisons of the young KP with older populations should all be percentages. It would be worth adding these in for greater clarity.

Thank you for catching this oversight. The text has been edited with the percentages to ensure consistency when communicating the data. 

Page 15. Line 222: change to ‘higher education levels’

Thank you, correction has been adopted. 

Page 19, line 312: review this line. The ‘however’ suggests that the second half of the sentence contradicts or is different to the first and this is not the case in this instance.

Thank you. The sentence has been revised for clarity: 

Finally, it is not currently possible to track and analyze the HIV epidemic among the adolescent population in general, and young KP specifically, because of limited health information systems.

---

## [Editor Report · Decision Letter 1]

15 Dec 2021

Young key populations left behind: The necessity for a targeted response in Mozambique

PONE-D-21-05957R1

Dear Dr. Boothe,

We’re pleased to inform you that your manuscript has been judged scientifically suitable for publication and will be formally accepted for publication once it meets all outstanding technical requirements.

Kind regards,

Mitzy Gafos

Guest Editor

PLOS ONE
---

## [Editor Report · Acceptance letter]

21 Dec 2021

PONE-D-21-05957R1 

Young key populations left behind: The necessity for a targeted response in Mozambique       

Dear Dr. Boothe:

I'm pleased to inform you that your manuscript has been deemed suitable for publication in PLOS ONE. Congratulations! Your manuscript is now with our production department. 

Kind regards, 

on behalf of

Dr. Mitzy Jane Gafos 

Guest Editor

PLOS ONE